# Analysis of Progression Time in Pancreatic Cancer including Carcinoma In Situ Based on Magnetic Resonance Cholangiopancreatography Findings

**DOI:** 10.3390/diagnostics11101858

**Published:** 2021-10-09

**Authors:** Kentaro Yamao, Masakatsu Tsurusaki, Kota Takashima, Hidekazu Tanaka, Akihiro Yoshida, Ayana Okamoto, Tomohiro Yamazaki, Shunsuke Omoto, Ken Kamata, Kosuke Minaga, Mamoru Takenaka, Takaaki Chikugo, Yasutaka Chiba, Tomohiro Watanabe, Masatoshi Kudo

**Affiliations:** 1Department of Gastroenterology and Hepatology, Kindai University Faculty of Medicine, Osaka-Sayama, Osaka 589-8511, Japan; g21001@edu.med.kindai.ac.jp (K.T.); atanakjp@gmail.com (H.T.); ackiy0409@gmail.com (A.Y.); a-o-k@mail.goo.ne.jp (A.O.); chochiko.4kg@gmail.com (T.Y.); shunsuke.oomoto@gmail.com (S.O.); ky11@leto.eonet.ne.jp (K.K.); kousukeminaga@yahoo.co.jp (K.M.); mamoxyo45@gmail.com (M.T.); tomohiro@med.kindai.ac.jp (T.W.); m-kudo@med.kindai.ac.jp (M.K.); 2Department of Diagnostic Radiology, Kindai University Faculty of Medicine, Osaka-Sayama, Osaka 589-8511, Japan; mtsuru@dk2.so-net.ne.jp; 3Department of Pathology, Kindai University Faculty of Medicine, Osaka-Sayama, Osaka 589-8511, Japan; tchikugo@mac.com; 4Clinical Research Center, Kindai University Hospital, Osaka-Sayama, Osaka 589-8511, Japan; chibay@med.kindai.ac.jp

**Keywords:** carcinoma in situ, computed tomography, early diagnosis, magnetic resonance cholangiopancreatography, natural history, pancreatic carcinoma

## Abstract

Background: Pancreatic cancer (PC) exhibits extremely rapid growth; however, it remains largely unknown whether the early stages of PC also exhibit rapid growth speed equivalent to advanced PC. This study aimed to investigate the natural history of early PCs through retrospectively assessing pre-diagnostic images. Methods: We examined the data of nine patients, including three patients with carcinoma in situ (CIS), who had undergone magnetic resonance cholangiopancreatography (MRCP) to detect solitary main pancreatic duct (MPD) stenosis >1 year before definitive PC diagnosis. We retrospectively analyzed the time to diagnosis and first-time tumor detection from the estimated time point of first-time MPD stenosis detection without tumor lesion. Results: The median tumor size at diagnosis and the first-time tumor detection size were 14 and 7.5 mm, respectively. The median time to diagnosis and first-time tumor detection were 26 and 49 months, respectively. Conclusions: No studies have investigated the PC history, especially that of early PCs, including CIS, based on the initial detection of MPD stenosis using MRCP. Assessment of a small number of patients showed that the time to progression can take several years in the early PC stages. Understanding this natural history is very important in the clinical setting.

## 1. Introduction

Pancreatic cancer (PC) displays a rapid tumor progression [1,2,3] and has the worst survival rate among the common types of cancer, as shown by the low 3-year survival rate (3%) in unresectable PC cases [4]. Indeed, almost all patients are diagnosed at an advanced stage. However, recent genetic and experimental study suggests that transformation of high-grade pancreatic intraepithelial neoplasia (PanIN), a pre-cancerous lesion of PC, into invasive PC required a time period ranging from 3 to 5 years [5]. A few cases have been published in which definite early PC had developed during a long observation period after the initial detection of main pancreatic duct (MPD) stenosis [6,7]. Although these case reports strongly suggested a long-term onset of early PC, estimation of the tumor progression time is very difficult because of the small number of reported patients being diagnosed with early PC. Therefore, little is known regarding the clinical features of early PCs and time to progression from carcinogenesis to invasive cancer. This limited knowledge concerning the natural history of early PCs may sometimes lead to misunderstanding of tumor progression time in PCs, especially those in the early stages, because image abnormalities highly suggestive of early PCs, such as MPD stenosis or a tiny tumor, may not be detected as morphological changes. Given that early PCs have a long-term prognosis after surgical intervention [8], understanding the natural history of early PCs is clinically very important.

Previous clinical studies have estimated the time to progression of PCs using pre-diagnostic images [1,9,10,11,12,13]. Most studies utilizing pre-diagnostic computed tomography (CT) scans have suggested that MPD abnormalities (dilation, stenosis, or interruption) and small nodules can be detected within 1 year before a definitive diagnosis of malignancy [9,10,11,12,13]. However, those studies had several limitations. The first issue was the tumor size. A PC with a tumor size >20 mm was considered to be an invasive tumor present at the time when the pre-diagnostic images were taken; therefore, these results cannot accurately reflect the time to progression from the early stage of PC. The second issue concerns the modality for assessment of MPD stenosis. Solitary MPD stenosis has been reported to be an essential secondary finding of early PCs [8], and magnetic resonance cholangiopancreatography (MRCP) is more sensitive than CT in detecting this abnormality [14]. Although some studies have shown that MPD stenosis was a preceding indicator of PCs, those data were obtained through analyzing CT images alone. This study aimed to assess pre-diagnostic images, including MRCP, that had been taken >1 year earlier. Moreover, we aimed to investigate the natural history of early PCs. We hypothesized that assessment of MPD stenosis based on more sensitive MRCP would facilitate determining a more accurate progression period of PCs than that reported in previous studies [1,9,10,11,12,13].

## 2. Materials and Methods

### 2.1. Patient Selection

We included patients who met all of the following criteria: (i) a diagnosis of PC based on pathological analysis (of surgical specimen or endoscopic ultrasound-guided fine needle aspiration [EUS-FNA]); (ii) MRCP findings indicated solitary MPD stenosis >1 year before the diagnosis; and (iii) previous history of various imaging (contrast-enhanced CT (CE-CT) and/or EUS) and follow-up MRCP examinations of pre-existing MPD stenosis. We excluded patients who met any of the following criteria: (i) a diagnosis of intraductal mucinous papillary carcinoma based on pathological analysis; (ii) no detection of MPD stenosis on MRCP; and (iii) a recurrent PC after surgery. Ethical approval for this retrospective study was granted by the relevant review boards of Kindai University Faculty of Medicine (registration number: R03-027).

### 2.2. Outcome Measurements and Definitions

We aimed to investigate the natural history of early PCs through retrospectively assessing pre-diagnostic images. Our primary objectives were to analyze time to diagnosis and identify first-time tumor detection. Our secondary objectives were to assess the clinical results, tumor size at first detection, and final tumor size at diagnosis. Time to diagnosis was measured from first-time detection of MPD stenosis without tumor lesion to diagnosis. First-time detection of MPD stenosis without tumor lesion was defined as the time point of the first MRCP-detected MPD stenosis without tumor lesions on the CT image. In patients who had undergone surgery, the day of diagnosis was determined as the day of surgery. In patients who did not undergo surgery because of tumor progression, the day of diagnosis was determined as the day of EUS-FNA performance. The time to first-time tumor detection was calculated from first-time detection of MPD stenosis without tumor lesion to first detection of the tumor lesion in any images. Image analyses of CE-CT and magnetic resonance imaging (MRI) were interpreted independently, but not blindly, by two reviewers: a radiologist (M.T., with 25 years of image interpretation experience) and a gastroenterologist (K.Y., with 18 years of clinical experience in pancreatobiliary disorders). One patient with carcinoma in situ (CIS) did not show tumor lesions in any images and was treated as censored in the analysis of time to first tumor detection. We considered CIS as a high-grade PanIN [15,16], which was treated as a cancerous lesion in this study. The tumor size on EUS was also interpreted by two reviewers: two gastroenterologists (K.Y. and K.M., with 17 years of clinical experience with pancreatobiliary disorders). If the two reviewers did not agree on the tumor size, a consensus was reached following discussion between the two reviewers. The final tumor size was determined through assessment of pathological specimens in patients who had undergone surgery without neoadjuvant chemotherapy (NAC) or determined using the final EUS images in patients who had undergone surgery post-NAC and in patients who had not undergone surgery because of tumor progression. The patients with CIS were considered to have a 0 mm sized tumor in the final tumor size analysis. Final EUS was defined through a EUS examination just before NAC or at the time of EUS-FNA in patients who had not undergone surgery.

### 2.3. Statistical Analyses

Medians and standard deviations were used to describe continuous variables, and percentages were used for categorical variables. The time to diagnosis and first-time tumor detection were calculated using Kaplan–Meier analysis. Statistical analyses were performed using GraphPad Prism 8 (GraphPad Software Inc., San Diego, CA, USA) software.

## 3. Results

### 3.1. Patient Selection

Through a retrospective analysis of patient medical records, we identified 1112 patients who had been diagnosed with PC at our institution between January 2004 and December 2020. Of these, we selected 88 patients with multiple images, including MRCP, taken over the 1-year course. We excluded 79 patients; among them, 74 patients had been diagnosed with PC within 1 year after MRCP and five patients had not exhibited MPD stenosis in the previous MRCP. Finally, nine patients who met the inclusion criteria were included in the study (Figure 1).

### 3.2. Patient Characteristics

The patient characteristics are presented in Table 1. The reasons for performing the first MRCP were as follows: follow-up examination because of a history of idiopathic acute pancreatitis (n = 3), further examination of MPD stenosis detected using other images (n = 3), further examination of acute pancreatitis (n = 2; idiopathic (n = 1) and alcohol-related (n = 1)), and further examination of pancreatic cysts (n = 1). The reasons for follow-up examination after the first MRCP were as follows: MPD stenosis, in which findings had been tentatively diagnosed as chronic pancreatitis (CP, n = 4); no confirmation of malignancy according to pancreatic juice cytology results despite a suspicion of malignancy (n = 2); MPD stenosis detected in retrospective image analyses, whereas no abnormalities had been noted earlier (n = 2); and MPD abnormality tentatively diagnosed as an intraductal papillary mucinous neoplasm (IPMN) (n = 1). Five patients had a history of acute pancreatitis, although no patients had calcification consistent with MPD stenosis as in suspected CP.

### 3.3. Diagnosis Assessment Results and First-Time Tumor Detection

Diagnosis assessment results are presented in Table 2, Appendix A, and Figure 2A. Pathological diagnosis was confirmed through surgical specimen analysis in seven patients and through EUS-FNA in two patients who were unsuitable surgical candidates because of distant metastasis and underlying diseases. The final tumor size was assessed through surgical specimen analysis in five patients, with 0 mm in patients with CIS in three patients, 3 mm in one patient, and 20 mm in one patient. The final tumor sizes determined using final EUS were 14 mm and 36 mm in patients who received NAC before surgery, and 16 mm and 28 mm in those who could not receive surgery. The median final tumor size was 14 mm, and the median time to diagnosis, which was calculated from first-time detection of MPD stenosis without tumor lesion (i.e., initial detection of MPD stenosis to diagnosis), was 26 months (range, 14–55 months; Figure 2A). The first tumor detection assessment results are presented in Table 2, Appendix A, and Figure 2B. The median tumor size in the image at first detection was 7.5 mm (range, 4–12 mm; three patients with CIS were treated as censored owing to no detection in any images). In addition, the first-time tumor detecting imaging modalities were CE-CT and EUS in four and two patients, respectively; however, tumor lesions were not detected in any of the three patients with CIS. The median time to first tumor detection was 49 months according to the Kaplan–Meier analysis (Figure 2B). Thus, the time to first tumor detection was longer than the time to diagnosis. This discrepancy can be explained by the presence of three patients with CIS, for whom tumor detection was not possible; therefore, these patients were treated as censored for the statistical analyses. We summarized representative cases in the following Figure 3, Figure 4, Figure 5 and Figure 6 (please see the figure legends for clinical information).

**Figure d64e387:**
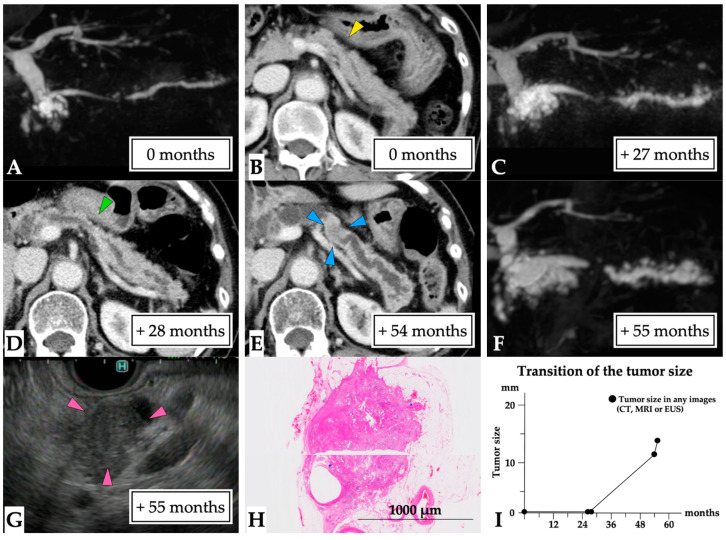


## 4. Discussion

In this study, we assessed numerous pre-diagnostic images, including MRCP, which had been undertaken >1 year earlier, and we clarified tumor progression time concerning early PCs. Our findings indicated that the median time to first detection of a tumor lesion was 49 months, with a median detected tumor diameter of 7.5 mm. Moreover, the median time to the diagnosis of PC was 26 months, with a median final tumor diameter of 14 mm. Thus, our data, which included the time point of first-time detection of MPD stenosis without tumor lesion, clearly showed that progression time concerning early PC is lower than previously reported [1,2,3,9,10,11,12,13]. Notably, the median time to tumor detection was longer than the median time to diagnosis. We attributed this finding to the presence of three patients with CIS for whom tumor detection was not possible and who were censored in terms of the statistical analyses.

Previous studies have investigated the time to progression of PCs using a doubling time (DT) calculation when analyzing pre-diagnostic images. Two studies [1,2] have reported that the DTs of PC were earlier than those in other types of cancers [17,18]. Moreover, almost all PCs have been diagnosed at an advanced stage [4], and it has previously been well established that PC is one of the fastest progressing cancers. Several studies have estimated the progression time of PCs using prior imaging findings in relation to the suspected presence of a tumor [1,9,10,11,12,13,19]; most of these studies [9,11,13,19] have reported that MPD abnormalities (stenosis, dilation, and interruption) were common findings appearing at 11–36 months before diagnosis. Interestingly, MPD abnormalities were considered to be the earliest pre-diagnostic findings in some studies [9,10,13]. Along with MPD abnormalities, focal hypoattenuating lesions [10,12,13,19], distal parenchymal atrophy [1,10,19], and loss of fatty marbling [9,11] have also been identified in pre-diagnostic images within 2 years before diagnosis. In contrast to these previous studies confirming the rapid growth speed of PCs, the times to diagnosis and to first-time tumor detection in our study were 26 and 49 months, respectively. Thus, the tumor progression time was considerably longer in our study than that previously reported. This difference can be partially explained by our MRCP-based assessment.

In our analysis, we included three patients with CIS in whom tumor lesions could not be detected in any of the images, whereas the median tumor size was >20 mm in previous studies [1,9,10,11,12,13,19]. To investigate the natural history of early PCs, the data of patients with CIS should be analyzed. The time to first tumor detection was longer in our study than that reported in previous studies. This finding may have been attributed to differences in the assessing imaging modalities used to detect MPD abnormalities. In previous studies, MPD abnormalities were assessed using CT scans [9,10,11,12,13,19], which have a reported lower sensitivity than that of MRCP [14]. In our study, we defined first-time detection of MPD stenosis without tumor lesion as the time point of the first MRCP-detected MPD stenosis, not CT-detected MPD stenosis. This definition, based on MRCP-detected MPD stenosis, facilitated a more accurate evaluation of the time to progression of PCs. However, it should be noted that a definition of first-time detection of MPD stenosis without tumor lesion based on the appearance of MPD stenosis may not be adequate because PCs also arise from the branch pancreatic duct [20]. Furthermore, a genetic and experimental study suggested that transformation of high-grade PanIN into invasive PC required a time period ranging from 3 to 5 years [5]. Therefore, the actual time to first tumor detection may be longer than that shown in our results.

Four of nine patients had been tentatively diagnosed with CP after the first MRCP, despite no detection of pancreatic calcification in the parenchyma. An image diagnosis of MPD stenosis without pancreatic calcification and tumor lesions is challenging, whether malignant or benign. According to the Japanese diagnostic criteria [21], pancreatic calcification is not an essential finding for a CP diagnosis. In our study, pancreatic calcification was not detected in patients with MPD stenosis. Previous studies have reported very low numbers of patients with PC and pancreatic calcification (0–5.3%) [22,23,24]; therefore, caution is needed when diagnosing patients with MPD stenosis without pancreatic calcification or tumor lesions. In this regard, recent studies have reported that an assessment of partial pancreatic parenchymal atrophy around small PCs is very useful for the diagnosis of small PCs [23,24,25,26,27,28]. Indeed, partial parenchymal atrophy was detected in our study patients with 14 mm and 36 mm lesions (Cases 5 and 9 in Appendix A) on the initial CT; therefore, the presence of partial pancreatic parenchymal atrophy could be a significant finding suggestive of PC.

Five of the nine patients had a history of acute pancreatitis. Previous studies have investigated the relationship between acute pancreatitis and PC [29,30,31,32,33,34]. The number of patients diagnosed with PC following acute pancreatitis has been reported to range from 0.7% to 12.4% [29,30,31,32,33,34]. The mechanism of acute pancreatitis associated with PC is attributed to obstruction of the MPD followed by release of pancreatic enzymes into the pancreatic parenchyma [33,34,35,36]. Furthermore, non-gallstone [33,37] and idiopathic acute pancreatitis [29,38] are reported risk factors for PC. It should be noted, nevertheless, that these previous studies addressing the incidence of acute pancreatitis before the diagnosis of PC have not assessed the presence or absence of MPD stenosis using CT or MRCP; thus, the first detection of MPD stenosis could not be set as the time of first-time detection of MPD stenosis without tumor lesion [29,30,31,32,33,34,35,36]. However, MPD stenosis was detected using MRCP in all our study patients with a history of acute pancreatitis. Therefore, these data strongly suggested that performing MRCP in patients with non-gallstone or idiopathic acute pancreatitis could help diagnose early PC. Future prospective studies with a large number of patients with acute pancreatitis are required to confirm this finding.

Our study had the following strengths. First, we defined the starting point for assessing the progression time as first-time detection of MPD stenosis without tumor lesion. MRCP has the highest sensitivity modality to assess MPD abnormality, which facilitated a more accurate assessment of the progression period of PCs than that reported in previous studies [1,9,10,11,12,13,19]. Second, we evaluated several patients with small tumors, including three patients with CIS, which differed from previous studies [9,10,11,12,13]. The larger number of patients with early PCs rather than those with advanced PCs provided a basis for us to conclude that the progression speed of early PCs is slower that that reported in previous studies.

Our study had some limitations. First, we enrolled a very limited number of patients because of the strict patient criteria. Second, this was a retrospective study, and the time interval and type of images were heterogeneous. Third, there was no confirmation of malignancy at the first MRCP. However, it should be considered that, in each case, the site of MPD stenosis corresponded with the diagnosed PC, and that MPD stenosis was in the initial stage of PC. Fourth, the images used to calculate tumor diameter were heterogeneous as we assessed the tumors using various modalities (CT, MRI, and EUS). For calculating the final tumor size, we need to assess the size using a resected specimen. Unfortunately, four out of nine patients underwent surgery after NAC or did not receive surgery because of tumor progression. EUS images were used to determine the tumor size in EUS images, as EUS was more accurate than the other imaging modalities (abdominal ultrasonography, CT, and MRI), especially for smaller lesions [39,40]. Tumor size was also independently calculated by a radiologist and two gastroenterologists for more accurate assessment. Finally, high-grade PanIN was treated as a cancerous lesion in our study, whereas this lesion has been treated as a precursor to PC outside of East Asia [41]. Thus, estimation of tumor progression time may be needed separately for patients with high-grade PanIN alone or with invasive PCs. Collectively, heterogenous populations with PC were analyzed in our study and, therefore, a prospective study including a large number of patients exhibiting MPD stenosis on MRCP is absolutely required to draw a conclusion that the progression time is much slower than that previously reported.

## 5. Conclusions

Few studies have investigated the natural history of PCs, especially early PCs, based on the findings of MPD stenosis detected using MRCP. In our study, although a small number of patients were assessed, the time to progression in the early stage of PCs may be much longer than that of our estimation. Patients with MPD stenosis without pancreatic calcification and tumor lesions may possibly have early PCs. In addition, patients with non-gall stone or idiopathic acute pancreatitis may require receiving MRCP to exclude early PCs. It needs to be emphasized that patients displaying MPD stenosis on MRCP require careful follow-up examination as they were considered as having a high risk for PC development. An understanding of this natural history of early PC aids clinical practice and improvement of the prognosis of this malignancy.

## Figures and Tables

**Figure 1 diagnostics-11-01858-f001:**
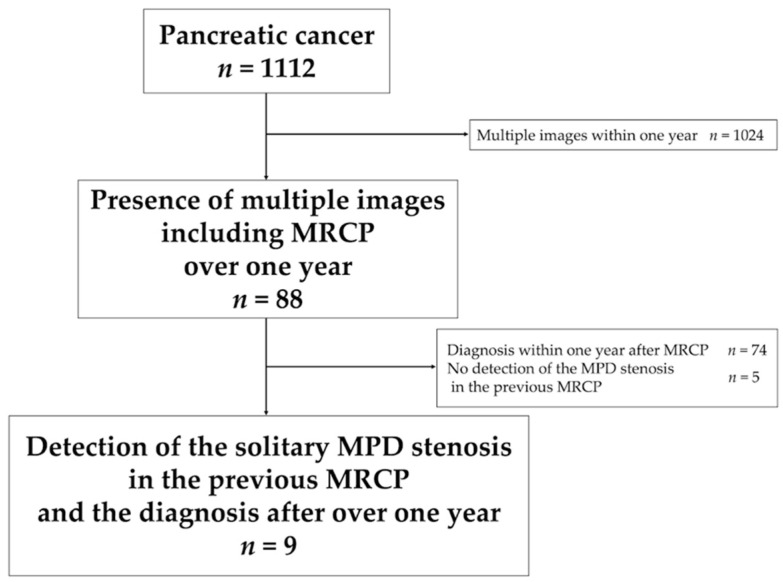
Flow-chart detailing the patient selection process. Abbreviations: MPD, main pancreatic duct; MRCP, magnetic resonance cholangiopancreatography.

**Figure 2 diagnostics-11-01858-f002:**
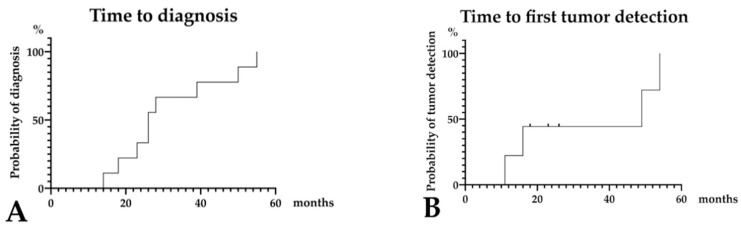
Kaplan–Meier curves. (**A**): Time to diagnosis (the median time to diagnosis was 26 months), (**B**): time to first-time tumor detection (the median first-time tumor detection was 49 months after three patients with CIS were censored). Abbreviation: CIS, carcinoma in situ.

**Figure 3 diagnostics-11-01858-f003:**
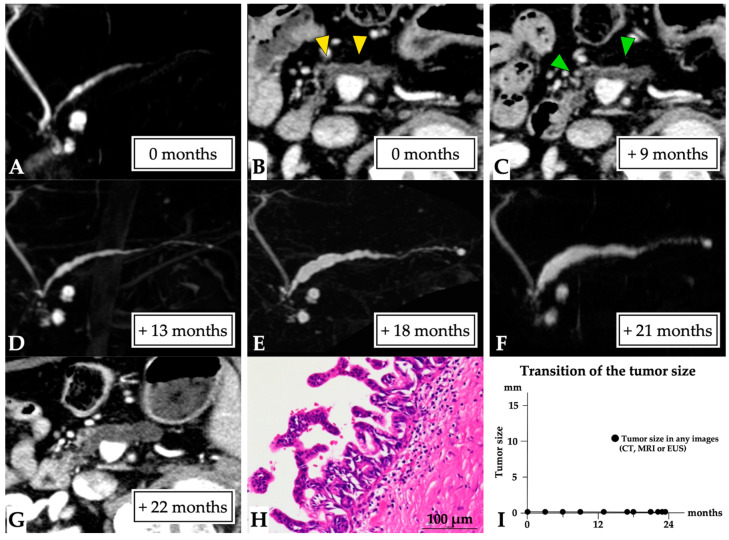
**Carcinoma in situ (0 mm in size) diagnosed over a 23-month observation period (Case 1 in Appendix A).** A 79-year-old man was referred to our hospital for further examination concerning pancreatic head cysts that had been observed during abdominal ultrasonography screening. MRCP showed MPD stenosis in the pancreatic head with slight distal MPD dilation, in addition to the detected cysts ((**A**): first-time MRCP). CE-CT (**B**) and EUS images also revealed MPD dilation (yellow arrow head), but not a tumor lesion around the MPD stenosis; however, severe distal pancreatic parenchymal atrophy was detected. Therefore, these lesions were initially diagnosed as branch duct IPMN, and careful follow-up examination was required. Subsequently, although no changes to the pancreatic cysts and MPD abnormality findings were observed on CT (green arrow head: MPD dilation, (**C**)) and MRCP (**D**), MPD stenosis, and distal MPD dilation were clearly observed on MRCP after 18 (**E**) and 21 months (**F**). However, no tumor lesions were detected on CT (**G**) or EUS images. Endoscopic retrograde cholangiopancreatography was performed as malignancy was suspected. Pancreatic juice cytology was not performed because of failure to cannulate into the MPD. Despite the lack of a definitive diagnosis, the lesion was resected after 23 months because of possible malignancy. The final diagnosis was high-grade PanIN of the MPD, which had only spread in the MPD in the pancreatic head (Tis N0 M0, stage 0, final tumor size: 0 mm (CIS), (**H**)), along with retention cysts in the pancreatic head. This patient was diagnosed at 23 months after the first MRCP (**I**). Abbreviations: CIS, carcinoma in situ; EUS, endoscopic ultrasound; CE, contrast-enhanced; CT, computed tomography; IPMN, intraductal papillary mucinous neoplasm; MPD, main pancreatic duct; MRCP, magnetic resonance cholangiopancreatography; PanIN, pancreatic intraepithelial neoplasia.

**Figure 4 diagnostics-11-01858-f004:**
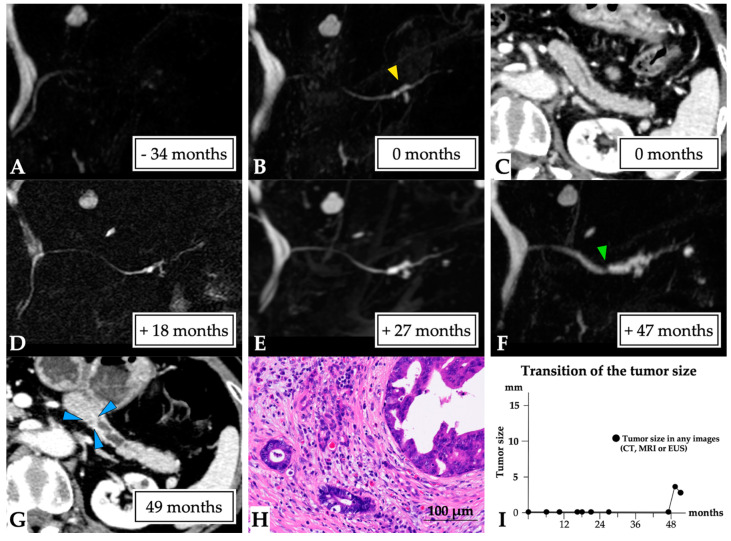
**A 3 mm lesion over a 50-month observation period (Case 4 in Appendix A).** The case of a 73-year-old woman who had a history of idiopathic acute pancreatitis at 4 years prior is presented. No MPD abnormalities were observed on MRCP (**A**); however, solitary MPD and branch pancreatic duct dilation were evident in the pancreatic tail (yellow arrow head, first-time MRCP, (**B**)), although no tumor lesions were detected on CT (**C**). The MPD abnormality gradually progressed (**D**,**E**), and MPD stenosis was detected at the beginning of solitary MPD dilation after 47 months (green arrow head, (**F**)). No tumor lesion was observed on CT or EUS images. ERCP detected localized irregular stenosis with suspected malignancy by pancreatic juice cytology findings. In addition, a tiny lesion was detected on CT after 49 months (blue arrow head, (**G**)). Therefore, surgery was undertaken after 50 months for a suspected small PC. The final diagnosis was the presence of a 3 mm invasive PC in the pancreatic tail (T1 N0 M0, stage IA, final tumor size: 3 mm, (**H**)). This patient was diagnosed at 50 months after the first MRCP (**I**). Abbreviations: EUS, endoscopic ultrasound; CT, computed tomography; IPMN, intraductal papillary mucinous neoplasm; MPD, main pancreatic duct; MRCP, magnetic resonance cholangiopancreatography; MRI, magnetic resonance imaging; PC, pancreatic cancer.

**Figure 5 diagnostics-11-01858-f005:**
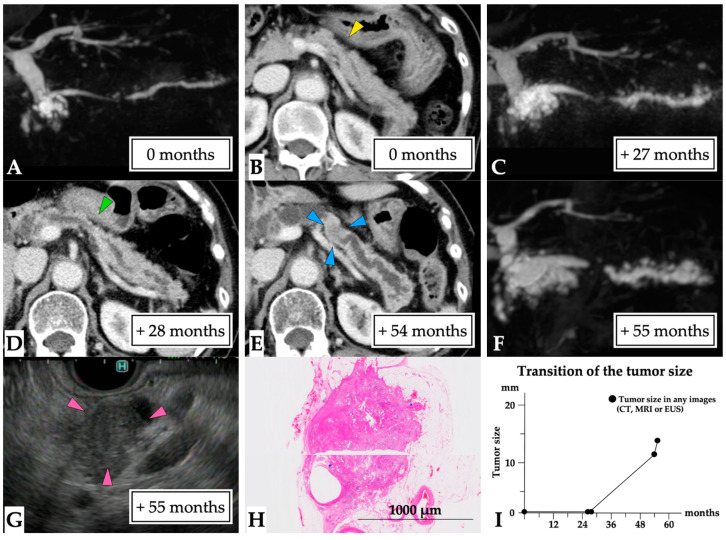
**A 14 mm lesion over a 55-month observation period (Case 5 in Appendix A).** The case of a 75-year-old woman who had experienced idiopathic acute pancreatitis 7 months prior is presented. MRCP findings revealed MPD stenosis in the pancreatic body accompanied by distal MPD dilation and multilocular cysts in the pancreatic head (first-time MRCP, (**A**)). CE-CT scan images failed to detect a tumor lesion, although partial parenchymal atrophy consistent with MPD stenosis was observed (yellow arrow head, (**B**)); therefore, MPD stenosis was treated as chronic pancreatitis despite the absence of pancreatic stones. MRCP performed 27 months later showed progression of distal MPD dilation (**C**), although no tumor lesion was observed on CE-CT scan images (green arrow head: partial parenchymal atrophy, (**D**)). Finally, CE-CT scan performed 54 months later indicated the presence of a tumor lesion (blue arrow head, (**E**)). In addition, MRCP detected further MPD dilation (**F**) and the patient was referred to our hospital. EUS scans performed 55 months later showed a 14 mm diameter tumor (pink arrow head, (**G**)). EUS-FNA was performed, and the pathological diagnosis was adenocarcinoma (final tumor size: 14 mm). Therefore, distal pancreatectomy was performed for PC post-NAC. The final diagnosis was the presence of a 12 mm invasive nodule in the pancreatic body (T1 N1 M0, stage IIB, (**H**)). This patient was diagnosed 55 months after the first MRCP (**I**). Abbreviations: EUS, endoscopic ultrasound; CE-CT; contrast-enhanced computed tomography; CT, computed tomography; FNA, fine needle aspiration; IPMN, intraductal papillary mucinous neoplasm; MPD, main pancreatic duct; MRCP, magnetic resonance cholangiopancreatography; MRI, magnetic resonance imaging; NAC, neoadjuvant chemotherapy; PC, pancreatic cancer.

**Figure 6 diagnostics-11-01858-f006:**
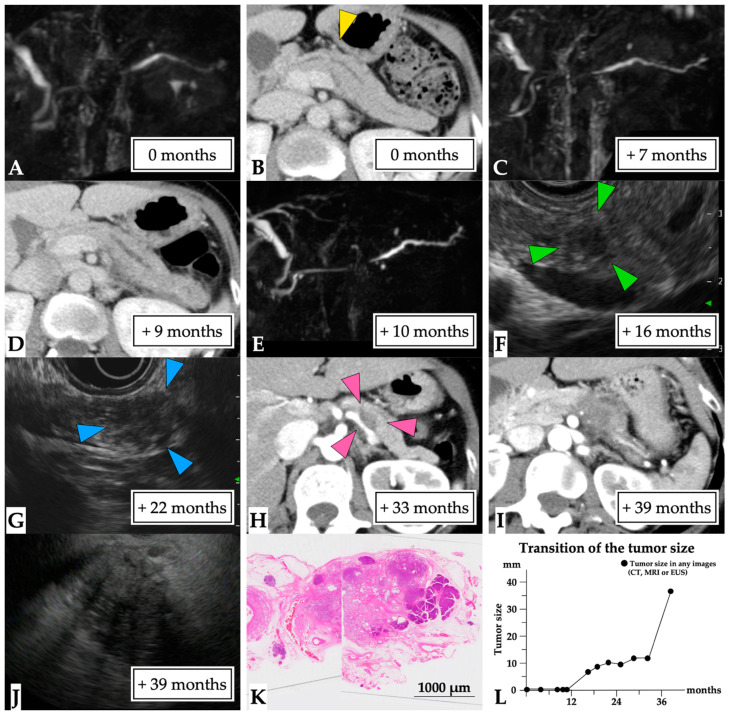
**A 36 mm lesion over a 39-month observation period (Case 9 in Appendix A).** A 49-year-old woman presented with idiopathic acute pancreatitis and underwent MRCP, which showed slight MPD stenosis accompanied by distal MPD dilation (first-time MRCP, (**A**)). CE-CT scans detected a partial MPD stenosis without tumor lesion or pancreatic stone (yellow arrow head, (**B**)), and the patient was diagnosed with chronic pancreatitis. Subsequently, a recurrent attack of pancreatitis occurred, and MPD stenosis and distal MPD dilation were clearly observed on MRCP and CT findings (**C–E**). An EUS scan performed after 16 months detected a 7 mm diameter tumor (green arrow head, (**F**)). However, EUS-FNA could not be performed because the endoscopist failed to recognize the tumor lesion. Therefore, ERCP was performed to determine a diagnosis of malignancy after 19 months. However, pancreatic juice cytology findings did not indicate malignancy. Subsequently, the tumor lesion was observed to have gradually progressed on EUS (blue arrow head, (**G**)) and CT (pink arrow head, (**H**)) images. This patient was referred to our hospital after 39 months following clear detection of the tumor (**I**). EUS-FNA was performed on the 36 mm diameter tumor (**J**), and the pathological diagnosis was adenocarcinoma (final tumor size, 36 mm). Therefore, distal pancreatectomy was performed for PC after NAC. The final diagnosis was the presence of a 30 mm invasive nodule in the pancreatic body (T3 N1 M0, stage IIB, (**K**)). This patient was diagnosed at 39 months after the first MRCP (**L**). Abbreviations: EUS, endoscopic ultrasound; CE-CT; contrast-enhanced computed tomography; CT, computed tomography; FNA, fine needle aspiration; IPMN, intraductal papillary mucinous neoplasm; MPD, main pancreatic duct; MRCP, magnetic resonance cholangiopancreatography; NAC, neoadjuvant chemotherapy; PC, pancreatic cancer.

**Table 1 diagnostics-11-01858-t001:** Patient demographic and clinical characteristics.

Age, median (range)	75 (49–84)
Sex, n (%)	
Male	6 (66.7)
Female	3 (33.3)
Main location of the lesion, n (%)	
Head/Body/Tail	1 (11.1)
Body	4 (44.4)
Tail	4 (44.4)
Reasons for first-time MRCP	
Follow-up post-acute pancreatitis	3 (33.3)
Examination of MPD stenosis	3 (33.3)
Examination of acute pancreatitis	2 (22.2)
Examination of pancreatic cysts	1 (11.1)
Reason for follow-up post-first MRCP, n (%)	
Tentative diagnosis of chronic pancreatitis	4 (44.4)
No evidence in pancreatic juice cytology despite a strong suspicion of pancreatic cancer	2 (22.2)
No abnormality noted *	2 (22.2)
Tentative diagnosis of IPMN	1 (11.1)
History of acute pancreatitis, n (%)	5 (55.6)
Calcification consistent with MPD stenosis, n (%)	0 (0)

* Detected following retrospective assessment. Abbreviations: IPMN, intraductal papillary mucinous neoplasm; MPD, main pancreatic duct; MRCP, magnetic resonance cholangiopancreatography.

**Table 2 diagnostics-11-01858-t002:** Diagnosis and first-time tumor detection.

Confirmation of pathological diagnosis, n (%)	
Surgery without NAC *	5 (55.6)
Surgery after NAC	2 (22.2)
EUS-FNA ^†^	2 (22.2)
Final tumor size, n (%) ^‡^	
0 mm ^§^	3 (33.3)
1–10 mm	1 (11.1)
11–20 mm	3 (33.3)
≥21 mm	2 (22.2)
First-time tumor detecting image modality	
CT, n (%)	4 (44.4)
No detection in any images, n (%) ^||^	3 (33.3)
EUS, n (%)	2 (22.2)

* Neoadjuvant chemotherapy, ^†^ inoperable patients, ^‡^ four patients assessed using EUS images, ^§^ CIS, ^||^ all cases were patients with CIS. Abbreviations: CIS, carcinoma in situ; CT, computed tomography; EUS, endoscopic ultrasound; EUS-FNA, endoscopic ultrasound-guided fine needle aspiration; NAC, neoadjuvant chemotherapy.

## Data Availability

The data presented in this study are available on request from the corresponding author. The data are not publicly available owing to privacy.

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
