# Peer review of "Analysis of Progression Time in Pancreatic Cancer including Carcinoma In Situ Based on Magnetic Resonance Cholangiopancreatography Findings"

_diagnostics, 2021, doi:10.3390/diagnostics11101858_

Round 1

Reviewer 1 Report

The authors investigated the natural history of early pancreatic cancer through retrospectively assessing prediagnostic images in nine patients. They concluded that time to progression can take several years in the early stages of pancreatic cancer.

I read this study again and unfortunately I do not see any significant improvement. The authors improved some technical details but all important issues still remain the same.

In their response they skipped one of the biggest limitation of this study – a sample size: The authors made their conclusions based on only 9 patients. This is too weak to draw any serious conclusion and to be published in any serious journal. The authors should continue and include more patients.

Of course I understand that ‘’tumor size in patients who did not undergo surgery cannot be calculated using a surgical specimen. In these patients, we assessed tumor size on EUS images’’ but this is two totally different approaches and results are not comparable. Tumor size measured by pathologist is never the same as tumor measured by imaging. In this case this is very important because for this particular study each millimeter is important, so you have two totally heterogenous groups

The source of bias regarding method of measurement and evidence or confirmation of malignancy at the first MRCP has not been improved. An explanation from the authors does not change the facts!

The authors should perform prospective study on larger sample size to reduce all mentioned important sources of bias. In this form, although it is interesting and well presented, this is not acceptable.

Author Response

Response: We would like to thank the reviewer for evaluating our manuscript and for the thoughtful comments. As the reviewer mentioned, the limited number of patients and the calculation method of the tumor size was a significant limitation of our study. We realize that we used an heterogeneous cohort of patients with PC and different methods to determine the tumor size. However, if we had excluded the patients with NAC or those who had not undergone surgery, the number of the remaining patients would have been only five. Therefore, we performed the analysis despite the presence of this heterogenicity. Despite such weakness, we believe that this study is worth publishing since our data clearly showed that progression of early PC is much longer than previously reported if we focus on MPD stenosis in MRCP examination. Thus, our study would make a contribution to the literature, as it highlights that we need to perform careful follow-up examination in patients displaying MPD stenosis on MRCP to detect early PC.

We agree with the reviewer that we need to perform a prospective study targeting a large number of patients to draw definitive conclusions. To avoid confusion and misunderstanding by the readership, we have revised the Discussion section and described the limitations of this study in a new paragraph as follows:

“Our study had some limitations. First, we enrolled a very limited number of patients because of the strict patient criteria. Second, this was a retrospective study, and the time interval and type of images were heterogeneous. Third, there was no confirmation of malignancy at the first MRCP. However, it should be considered that in each case the site of MPD stenosis corresponded with the diagnosed PC and that MPD stenosis was in the initial stage of PC. Fourth, the images used to calculate tumor diameter were heterogeneous as we assessed the tumors using various modalities (CT, MRI, and EUS). For calculating the final tumor size, we need to assess the size using resected specimen. Unfortunately, four out of nine patients underwent surgery after NAC or not received surgery because of tumor progression. EUS images were used to determine the tumor size in EUS images, as EUS examination was more accurate than the other imaging modalities (abdominal ultrasonography, CT, and MRI), especially for smaller lesions [39,40]. Tumor size was also independently calculated by a radiologist and two gastroenterologists for more accurate assessment. Finally, high-grade PanIN was treated as a cancerous lesion in our study, whereas this lesion has been treated as a precursor to PC outside of East Asia [41]. Thus, estimation of tumor progression time may be needed separately for patients with high-grade PanIN alone or with invasive PCs. Collectively, heterogenous populations with PC were analyzed in our study and, therefore, a prospective study including a large number of patients exhibiting MPD stenosis on MRCP is absolutely required to draw a conclusion that the progression time is much slower than that previously reported.” (Lines 663–680)

Reviewer 2 Report

In the present study, teh authors have attempted to assess pre-diagnostic images, including MRCP, that had been taken > 1 year earlier, and to investigate the natural history of early pancreatic cancer. They have hypothesized that assessment of MPD stenosis based on more sensitive MRCP would facilitate determining a more accurate progression period of PC than that reported in previous studies.

The authors have re-submitted their manuscript after previous review. They have greatly improved their work and the previous concerns have been dealt. Their work is interesting and it has merit for publication.

I have one final comment. Please highlight your findings and in particular, the potential uses of your approach to the prognosis, early diagnosis and therapy of PC. In addition, please comment on the time-dependent evolution of PC. What do we know on the natural history of the tumor from the point of the first cancer cells to the clinical presentation.

Author Response

Response: We would like to thank the reviewer for evaluating our manuscript and for the positive feedback. We have summarized and described our opinion regarding the potential uses of our approach to the prognosis, early diagnosis, and therapy of PC in the Conclusion section of the revised manuscript, as per the reviewer’s suggestion. The revised Conclusion section is as follows:

“Few studies have investigated the natural history of PCs, especially early PCs, based on the findings of MPD stenosis detected using MRCP. In our study, although a small number of patients were assessed, the time to progression in the early stage of PCs may be much longer than that of our estimation. Patients with MPD stenosis without pancreatic calcification and tumor lesions may have possibility of early PCs. In addition, patients with non-gall stone or idiopathic acute pancreatitis may require receiving MRCP to exclude early PCs. It needs to be emphasized that patients displaying MPD stenosis on MRCP require careful follow-up examination as they were considered as having a high risk for PC development. An understanding of this natural history of early PC aids clinical practice and improvement of prognosis of this malignancy.” (Lines 700–709)

The natural history of PC from the initiation of cancer cells to clinical presentation is beyond the scope of this manuscript. We have added more information concerning the progression time reported in the genetic and experimental study as follows:

“Furthermore, a genetic and experimental study suggested that transformation of high-grade PanIN into invasive PC required a time period ranging from 3 to 5 years [5].” (Lines 571–573)

Reviewer 3 Report

Early diagnosis is a crucial and still unsolved issue in pancreatic cancer. Although a small series of patients, due to a strict selection, this manuscript underlines the importance of MPD stenosis detected at MRCP; this MPD abnormality may establish even 49 months (median) before PC diagnosis. I have only two suggestions for the Authors:

  • The "representative case presentation" could be deleted from the results. These cases should be only described in the figure captions, and even more extensively than already done.
  • Please, deeply review the discussion to avoid redundant concepts.

Author Response

Response: We would like to thank the reviewer for evaluating our manuscript and for the positive feedback. Please note that we have removed the “representative case presentation” subsection and summarized the clinical information in the figure legends in the revised manuscript in accordance with the reviewer’s suggestion. In addition, we have changed the orders of figure compartments in each case. Moreover, we have carefully read the Discussion section again to remove the redundant portions and to shorten this section, as per the reviewer’s suggestion.

Round 2

Reviewer 1 Report

The authors provided response to my queries but nothing has changed. The main questions still remains:

  • The authors made their conclusions based on only 9 patients. This is too weak to draw any serious conclusion and to be published in any serious journal. The authors should continue and include more patients.
  • Tumor size measured by pathologist is never the same as tumor measured by imaging. In this case this is very important because for this particular study each millimeter is important, so you have two totally heterogenous groups
  • The source of bias regarding method of measurement and evidence or confirmation of malignancy at the first MRCP has not been improved. An explanation from the authors does not change the facts!

The authors should perform prospective study on larger sample size to reduce all mentioned important sources of bias. In this form, although it is interesting and well presented, this is not acceptable.

This manuscript is a resubmission of an earlier submission. The following is a list of the peer review reports and author responses from that submission.

Round 1

Reviewer 1 Report

The authors investigated the natural history of early pancreatic cancer through retrospectively assessing prediagnostic images in nine patients. They concluded that time to progression can take several years in the early stages of pancreatic cancer.

Although the study is well written I have some serious remarks in regards to methodology and study design:

As the author stated this condition has been investigated in several previous studies. The authors stated that all previously published studies had limitations such as tumor size or the modality for assessment of main pancreatic duct stenosis. The authors overcome these limitations but in my opinion in present study, although the study is well designed and written, there is one big limitation. The authors made their conclusions based on only 9 patients. This is too weak to draw any serious conclusion and to be published in any serious journal.

The authors stated that ''Time to first-time tumor detection was calculated from tumorigenesis to first detection of the tumor lesion in any images''. It is unclear how the tumorigenesis was determined. How is it even possible to detect tumorigenesis by using images?

Also the criteria for definition of tumor size were not unique. In some cases tumor size was calculated from pathological specimens and in some cases (the patients who did not receive surgery) from EUS images. This should be a source of significant bias.

Next the images used to calculate tumor diameter were heterogeneous (CT, MR or EUS). This should also be a source of significant bias.

Finally, the authors do not have evidence or confirmation of malignancy at the first MRCP so they speculated that MPD stenosis could be linked to tumorigenesis which could lead to an early stage of disease.

Reviewer 2 Report

The study proposed by the authors makes valuable contributions to find both appropriate methods for early identification of pancreatic cancer and to make the most accurate estimates of progression time.

Highlighting the clinical aspects of early pancreatic cancer is extremely important.

Although the authors evaluated a long time interval (January 2004 and December 2020), based on the inclusion and exclusion criteria they identified only nine cases with MRCP having indicated solitary MPD stenosis>1 year before diagnosis and having undergone various imaging and follow-up MRCP studies of pre-existing MPD stenosis.

These case selection principles are correctly established both clinically and in terms of the analysis of the progression time of pancreatic cancer.

The presentation of retrograde endoscopic cholangiopancreatography (ERCP) in dynamics provides an assessment of the evolution of PC.

MRCP showed MPD stenosis in the pancreatic head with slight distal MPD dilation (first-time evaluation), compared with CT images that do not indicate the presence of a tumor lesion. Presentation of these aspects are of real clinical benefit and provide a correct view of the time of disease progression.

Through the design of the study and the presentation of the results, the paper makes remarkable contributions to the literature. The results are supported by remarkable discussions based on a recent bibliography.

Reviewer 3 Report

In the present manuscript the authors have investigated the possible time-dependent progression of pancreatic cancer by using imaging methods. In particular, their main idea was to identify patterns that would be able to detect pancreatic cancer early in its developing time-line, before its onset.

The idea per se is very interesting, yet the work has some flaws which should be addressed.

First of all there are some ethical issues which the authors do not address. They do not refer to a bioethics approval for their study. Please add it, or explain why this is not mentioned. In addition, I would suggest that the authors elaborate on the general ethical issue of their study, since in order to be able to draw conclusions from such studies, it is sometimes necessary for the disease to progress without treatment. How would they handle such an issue. This should be added as a separate paragraph in the "Discussion" section.

The authors report that their study concerned nine patients, yet there are only three figures (probably one for each patient). This brings me to my next comment, which is that in each figure legend the authors refer to "patients", yet from my understanding each figure presents one patient. This is unclear, they should clarify, as well as they should present all nine patients separately.

The histochemical subfigure in each figure is absolutely not clear. They should describe (and most importantly highlight, show) what we are seeing, which month, which patient etc.

They also should present the diagram of  tumor size for all patients included in their study, as well as combine their results and present the mean tumor progression, with standard deviations. In addition, since they have measured the size of the tumor for each time point, they should statistically report the significance of their findings. How is it to understand if their approach really can predict the progression of a tumor from initial imaging estimations?

If they are reporting the medians then the min and max are required. Standard deviations go with mean values, which they should use.

In addition, the authors have reported on this topic with their previous manuscript "Partial Pancreatic Parenchymal Atrophy Is a New Specific Finding to Diagnose Small Pancreatic Cancer (≤10 mm) Including Carcinoma in Situ: Comparison with Localized Benign Main Pancreatic Duct Stenosis Patients". What is the additional knoledge added with the present work?

Overall, although their study and concept is interesting and it could help to understand the dynamics of tumor progression, the presentation of their results lacks clarity and consistency.